# Electric vehicles in Danish Municipalities: An Understanding of Motivations, Barriers, and the Future of Sustainable Mobility

Tessa Kate Anderson

National Space Institute, Danish Technical University (DTU) SPACE, Elektrovej, Building 328, Room 009, 2800 Kgs. Lyngby, Copenhagen, Denmark; teka@space.dtu.dk; Tel.: +45-53541281

**Abstract:** This paper explores the procurement, use, and experience of plug-in electric vehicles (PEVs) in Danish municipalities in relation to the notion of early adopters and socio-technical theory. Denmark has been one of the most ambitious countries in terms of electric vehicle adoption and use. This study used a combination of in-depth surveys and interviews with all 61 Danish municipalities on their fleet PEV experience and use. By building on the literature, the paper offers a deeper understanding of decision-making pathways for the procurement of PEVs. PEVs were found to be most suited to certain departments and the acceptance and uptake of PEVs was found to be complex and not straightforward.

**Keywords:** innovation; early adopters; Denmark; procurement; decision-making; fleet cars

## 1. Introduction

According to the International Energy Agency (IEA) 2016, Denmark's electric car stock targets are the most ambitious out of 14 countries currently with the highest proportion of plug-in electric vehicles (PEVs) [1]. Denmark intends to increase its private market share of PEVs to 9% of the overall car market. This is ambitious as in 2015, Denmark announced it would be increasing its vehicle car tax for electric vehicles in line with those of normal gasoline and diesel cars, with an incremental rise to 180% tax by 2020. The Danish government changed in 2015 and with it came increased revenue incentives (from the government), which included the increase in taxes on electric vehicles. It is unknown whether the implementation of this 180% rate will go ahead. However, what is clear is the number of PEVs registered has fallen from approximately 5000 in 2015 to 700 in 2017.

Like many other countries, Denmark has also launched extensive research programmes for battery and vehicle development and introduced field trials to test technology and explore mobility solutions and business models. The Danish government has initiated several programmes and projects since 2009, such as the pilot regions of electro-mobility and four so-called 'show cases for electro-mobility', which aim to help Denmark achieve its self-set goal of becoming a leading market for electric mobility. In addition, local governments have also become active in the promotion of electric mobility.

Municipality or local authority based adoption of PEVs (politics and economics) is seen as an important step for wider based dissemination of electric vehicles for private use. Local municipalities in Denmark have a significant influence on the dissemination of electric vehicles, on top of the national government policies. The municipality vehicles are often seen driving around the local area and have a high visibility. Dijik et al. (2013) [2] describes how fleet operators are emerging as a key force influencing the directions of PEV development and commercialization. Many local governments also have instruments to stimulate electric vehicles. The number of PEVs in Denmark increased tenfold since mid-2012 until mid-2014 [1]. In addition, the awareness of PEVs and local instruments to

stimulate PEVs changed a lot during this period. The following policy measures have a significant effect on the number of electric vehicles: Charging infrastructure in the public space and the subsidy for and the purchase of an electric vehicle [2].

All municipalities throughout Denmark have a climate change policy agenda. The procurement of PEVs aligns itself to this agenda and the promotion of sustainable transport. Between 2008 and 2015, the Danish Energy Agency (DEA) carried out a test car scheme for PEVs. The project was open to companies, associations, municipalities, and regions (https://ens.dk/ansvarsomraader/transport/alternative-drivmidler/forsoegsordning-elbiler, accessed June 2017). A large majority of municipalities took advantage of this project, which is reflected in the increased number of procured PEVs from 2008 to 2013.

This research builds on the findings from Sierzchula (2014) [3], who acknowledges that although studies have shown organizations to be major early adopters of electric vehicles, there has been little empirical data evidence for identification of the factors that influence fleet manager purchasing decisions. Previous research [4,5] was conducted before the electric vehicles became an increasing proportion of the overall sales in vehicles. Municipality owned or leased PEVs tend to be an attractive first entry point to the transport system, which makes it important to study the users related to this context.

Socio-technical research, which integrates technical aspects with social science or behavioural perspectives, has been put forward as an important approach to effectively decarbonize the transport system [1,6]. To complement other studies that have focused primarily on technical and financial barriers, for example, Kalhammer et al. (2007) [7] and Egbue and Long (2012) [8], this paper investigates the operational barriers; such operational barriers ultimately determine overall usage and influence user acceptance. The crucial factor determining the success or failure of a new technology is users' acceptance and willingness to use it [9]. User acceptance is difficult to anticipate and the attitude towards a vehicle technology differs between potential users and actual users [10].

This paper focuses specifically on full electric vehicles (plug-in electric vehicles, PEVs) rather than hybrid vehicles (battery electric vehicles, BEVs and also some PEVs) because hybrid vehicles rely on fossil fuels and since purchasing decisions relating to hybrids are likely to involve factors different to those connected with EVs [3,11]. The study aims to identify factors (political, economic, and environmental, for example), which have influenced the purchasing or leasing of electric vehicles, and the challenges or incentives for future use. It embarks on an attempt to unpick the rationale behind the decision-making pathways of electric vehicle uptake in Danish municipalities. By combining an understanding of municipality based operational barriers and users' perspectives, this paper contributes new findings to the growing empirical research on the early adopters of electric vehicles.

The structure of the papers is as follows: Section 2 provides an outline of the research questions this paper aims to address; Section 3 focuses on the current state of knowledge in the literature, Section 4 outlines the research design, Section 5 provides the analysis of the survey and interviews, Section 6 outlines the results and discussion, and, finally, Section 7 concludes the paper.

## 2. Research Questions

Research has identified several reasons why organizations are such good candidates for being early EV adopters, including:

- High vehicle purchase rates [1];
- intense usage [2];
- centralized refueling stations [4];
- limited decision makers [1,2,4]; and
- fleet managers have a better understanding of lifetime costs of a vehicle than private households [10,11].

The aim of this paper is to begin to understand the barriers and opportunities associated with early adopters.

- What were the most important factors that influenced fleet managers to choose (or not choose) electric vehicles?
- How can we begin to understand the evolution of these early adopters of EVs and what are the transition pathways and future scenarios?
- How can we build a better knowledge base for the challenges faced by both public/ government organisations?

The purpose of this study is to build on empirical evidence, which suggests fleet vehicles are early adopters of EVs. Figure 1 shows the theoretical framework for this study.

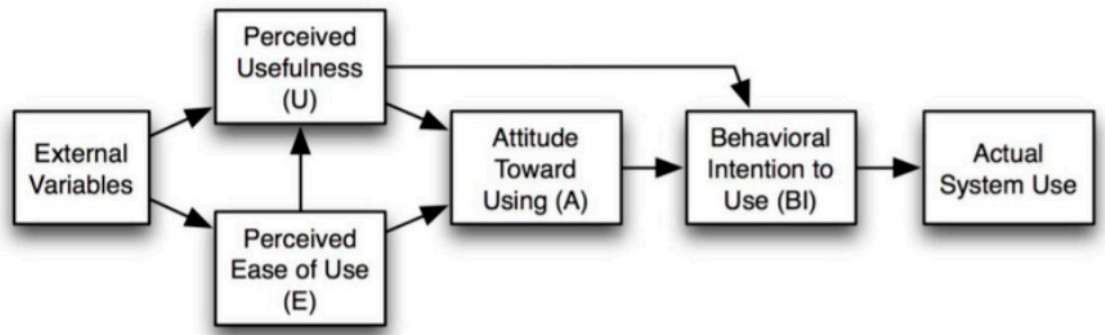

**Figure 1.** Technology acceptance model (TAM) created by Davis (1985) [9] with the purpose of explaining as well as predicting behavioural intentions of a person towards technological innovation.

The theoretical framework of the Technology acceptance model (TAM) has been used to gain a deeper understanding of the questionnaires and interviews. The TAM offers a method of understanding why users accept or reject technology. According to TAM, the users' experience of a technology system (in this instance, electric vehicles) is influenced directly or indirectly by the users' behavioural intentions, attitude, perceived usefulness, and perceived ease.

## 3. State of Knowledge

Research into the early adopters of electric vehicles is a relatively new phenomenon largely due to the newness of the vehicles themselves [3]. There has been a considerable amount of literature, which has focused on the market penetration of EVs in private households. However, there has been limited research into the motivations and barriers of the introduction of EVs in the commercial and local government sector [12–14].

Haller et al. (2007) [14] was one of the first research papers to explore the adoption of EVs in local government. The paper presented an assessment of a voluntary 10-year plan to convert to electric vehicles. There were three parts to the assessment, cost effectiveness analysis, implementation evaluation, and environmental outcome analysis. The study discussed the challenges in attaining uptake goals due to delays in anticipated grants and installing charging stations. The study looked at the programme half way through and highlights that the conversion was 50% complete, and whilst this was promising, it was difficult to attain cost savings and emission reductions. They identify limited funds to invest in infrastructure and vehicles to make significant progress.

Nesbitt and Sperling (2001) [4] analysed the decision processes of fleet managers concerning and discussing their implications in terms of the purchase of PEVs. The findings suggest that there are four decision styles: Autocratic, bureaucratic, hierarchic, and democratic, with an autocratic decision style pertaining to SMEs, and bureaucratic and hierarchic styles pertaining to large public or private organizations. Sierzchula (2014) [3] focused on the motivating factors and barriers

underlying the purchase of ECVs and established the importance of external considerations in the decision process. Kirk et al. (2014) [12] interviewed 17 stakeholders, including six company fleet managers, in a qualitative study in the UK regarding the market penetration of condensed natural gas commercial vehicles and found that possible motivating factors and barriers were fuel costs, refuelling infrastructure, vehicle purchase cost and residual value, the removal of the London congestion charge exemption, lack of knowledge regarding EV, and vehicle weight. Sierzchula (2014) [3] explored fleet managers' considerations regarding environmental impacts, financial incentives, and perceived operational ease with a qualitative study among 14 U.S. and Dutch organizations that adopted ECVs. The findings suggest the relevance of testing new technologies, receiving government grants, and improving the organization's public image, suggesting that attitudes towards technology and subjective norms play a role in the company's decision.

Most of the research has focused on consumer adoption of electric vehicles, and the adoption by public organizations has largely been overlooked. The first consumers of electric vehicles are of much interest in academic research due to their role in the successful introduction of new technology [15]. This is despite the fact that fleet operations provide the potential to address many consumer concerns expressed about electric vehicles, including addressing range limitation and recharging infrastructure, bridging information gaps about user behaviour and vehicle capability, and ensuring the reliability of the battery over time through warranty and maintenance arrangements.

Sierzchula (2014) [16] outlined several important theoretical concepts relevant to the emergence of EVs. The aim of this paper is to build on this literature review and discuss how specifically public agencies are highlighted as early adopters for the uptake of EVs. Technology and innovation literature identifies several important theoretical concepts and models that are particularly relevant and influential to the emergence of EVs. These include the technology acceptance model (TAM).

Kaplan (2016) [13] highlights that urban commercial fleet vehicles are considered good candidates for the early adoption of electric vehicles, largely due to their high mileage compared to household usage and possibility of frequent maintenance [17–21]. The literature on fleet vehicle procurement is scare and little is known about the behavioural framework underlying EV purchasing decisions as well as the internal and external barriers to EV adoption [22–24].

A study in the UK by Intelligent Leasing in 2015 (Intelligent Car Leasing, 2015) [25] conducted the single biggest study looking at local councils' adoption of electric vehicles. The results of the survey, where 433 local councils were surveyed as to the number of electric vehicles they owned or leased, revealed interesting trends across the UK. The survey was carried out under the Freedom of Information Act 2000 and highlighted that the highest number of EVs owned or leased by one council was 38 (Dundee City Council) followed by Lanarkshire and Glasgow. Scottish local councils were among the councils with the highest number of EVs. Wider implications suggest that whilst over one third of all local councils in the UK have at least one EV, this uptake would dwarf that of the private sector. It is unknown both in the UK and in Denmark how many private companies own or lease PEVs.

## 4. Research Design

The study chose to use a qualitative based methodology (questionnaires and interviews) because the low level of uptake of PEVS in the municipalities' precluded a large-scale statistical analysis. Data for this study came from a set of questionnaires sent to all the Danish municipalities (98) and a series of in-depth interviews with four municipalities. The questionnaire comprised of 15 questions. The interviews, which were conducted after the questionnaires, allowed for a more in depth and open discussion on PEVs, which was not possible through quantitative methods [26].

*4.1. Questionnaires*

Detailed questionnaires were emailed to all Danish municipalities between July 2015 and September 2015. Follow up phone calls were made to the municipalities who did not initially respond, with the final questionnaires being collected by the end of December 2015. The response rate was

high with a response from 61 municipalities (60% response rate). To find the appropriate contact at each municipality, the Danish Energy Agency (Energistyrelsen) provided the contact details for each municipality in Denmark. The representative completing the questionnaire was often the person responsible for procuring the PEVs. One of the main drawbacks in this instance was the person would not necessarily have direct experience of using the PEV and, therefore, the information would be anecdotal. The person answering the questionnaire could not always complete all the questions accurately. This was due to the complex and bureaucratic nature of the municipalities.

The data collected focused on the following themes:

- Overview of EVs in the municipality (number, type, age, government incentives);
- charging and range (satisfaction with charging infrastructure and range of the EVs); and
- the future of EVs in the municipality.

The questions were designed to give an overview of the past, present, and future status of PEVs in the municipality. All the questionnaires were in Danish. A pilot questionnaire was emailed to Roskilde municipality with a follow up meeting to discuss the questions and design and content.

### 4.2. Interviews

Interviews were conducted with four municipalities (Copenhagen, Roskilde, Frederikssund, and Aarhus). The municipalities were selected by asking the municipality if they would be willing to participate in a follow-up interview from the questionnaire. The interviews were semi structured and followed the questions asked in the questionnaire, but allowed for more in-depth answers. The interviews lasted approximately 60 min and were recorded and written notes where taken.

## 5. Analysis

### 5.1. Questionnaires

To analyse the results of the questionnaire, a mixture of descriptive statistics and non-parametric inferential statistics were used. The respondents of the questionnaire were able to write freely for certain questions and, therefore, content analysis was used. The survey data was analysed using exploratory data analysis (EDA) followed by the non-parametric inferential statistics. Data content analysis was also used on the answers of the questionnaire, which were freely written by the respondent.

### 5.2. Interviews

Interviews were conducted with representatives from four municipalities: Roskilde, Frederikssund, Aarhus, and Copenhagen. The municipalities were selected based on their cooperation for a follow up discussion about PEVs. To examine the interview data, content analysis was used, which is a reliable and systematic method of investigating this type of data [20] It involves developing textual categories according to specific rules and the subsequent codification of terms within the data. An emergent coding method [18] was employed, where researchers used a preliminary set of the data to independently identify keywords and phrases within the data, e.g., 'good value'. These factors were categorized into what is known in content analysis as textual categories. The textual categories were then checked by another researcher to make sure there was no anomalies ('intercoder reliability', Campbell et al., 2013). A coding scheme or checklist was developed to identify key words or phrases that were used and how many times they were used and with what strength.

## 6. Results and Discussion

This section has been subdivided into four main areas, which highlight the key themes, which were highlighted in both the surveys and interviews. The results try to unpick what this information can tell us about the roles of PEVs in Danish municipalities, what the barriers are to adoption, and thoughts of the future.

*6.1. Overall Situation and Management of PEVs in Danish Municipalities*

The number of PEVs in each municipality varied considerably. The largest urban areas in Denmark, Copenhagen (182), Aarhus (48), and Odense (108), had the most number of PEVS. Copenhagen's stock of PEVs was just under half of the total number of petrol and diesel cars (412). During the interview, it was evident the municipality was investing considerable amounts into both the procurement of PEVs and development of the charging infrastructure.

Of the 61 municipalities that responded to the survey, 18 did not have any PEVs. The majority of municipalities (31) had between one to 10 PEVs, and the remaining 12 municipalities had on average 23 PEVs. It is clear from the results that the urban municipalities have a higher procurement of PEVs. These municipalities tend to be larger in population and have a higher yearly transport budget.

The chain of information knowledge concerning EVs within the municipality showed a limited understanding of who was 'tracking' the EV progress. There was no common fleet management in place and individual 'departments' had autonomous purchasing power over the acquisition of electric vehicles. Often the questionnaire would be passed on to different people, and in some municipalities, the questionnaire was completed multiple times by different departments. Some of the municipalities refused to complete the questionnaire, citing that they were 'too busy', that they had 'answered many surveys', or the survey was 'too extensive'. The process of finding the most suitable person in the municipality was not straightforward and highlighted that each municipality managed the PEV fleet procurement process differently.

The majority of PEVs in Danish municipalities were procured after 2011, with a peak in 2012–2013. Figure 2 shows the dissemination of the results by year and by urban, country, and suburban municipalities. There are many 'semi-urban' municipalities that do not have any PEVs, compared to a high proportion of country-classified municipalities.

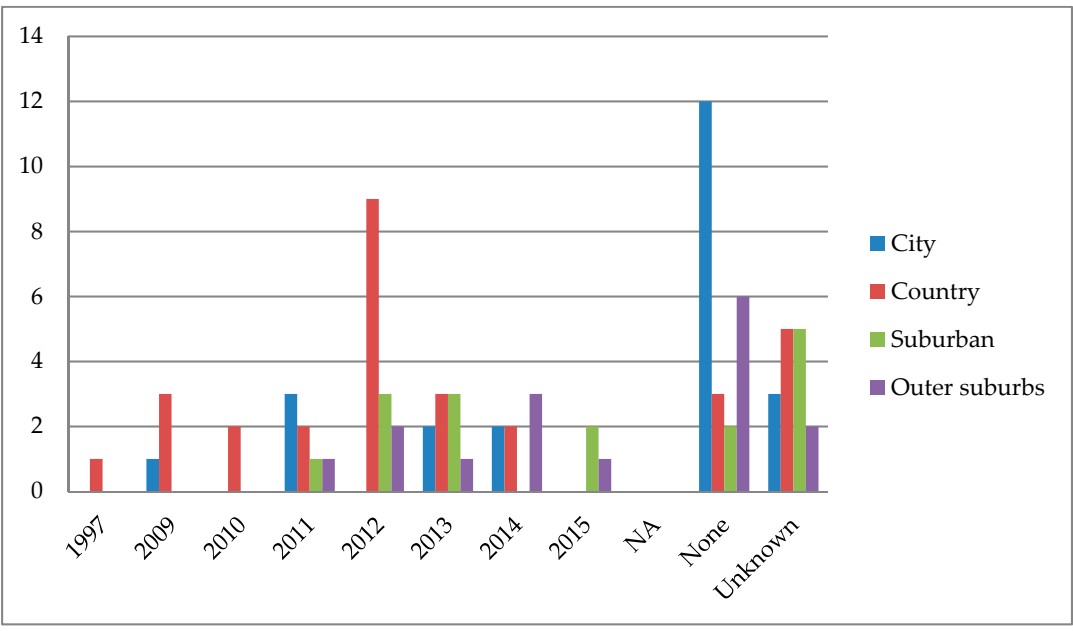

**Figure 2.** Graph to show the first year the municipality purchased electric vehicles.

*6.2. Reasons for Procurement*

The main reason for procuring PEVs was for environmental reasons, specifically reducing $CO_2$ emissions. Figure 3 shows some of the main reasons why the municipalities decided to procure the PEVs. Interestingly, the second biggest reason was a political decision. Interviews with the larger municipalities, such as Copenhagen and Aarhus, reinforced this result. The interview respondents highlighted how the decision to procure PEVs was a 'top-level management decision' usually in line

with the political policies of the city's future transport and environmental plans. Interestingly, financial motivations were lower down on the reasons for procuring PEVs, however, this could be masked by the project involvement.

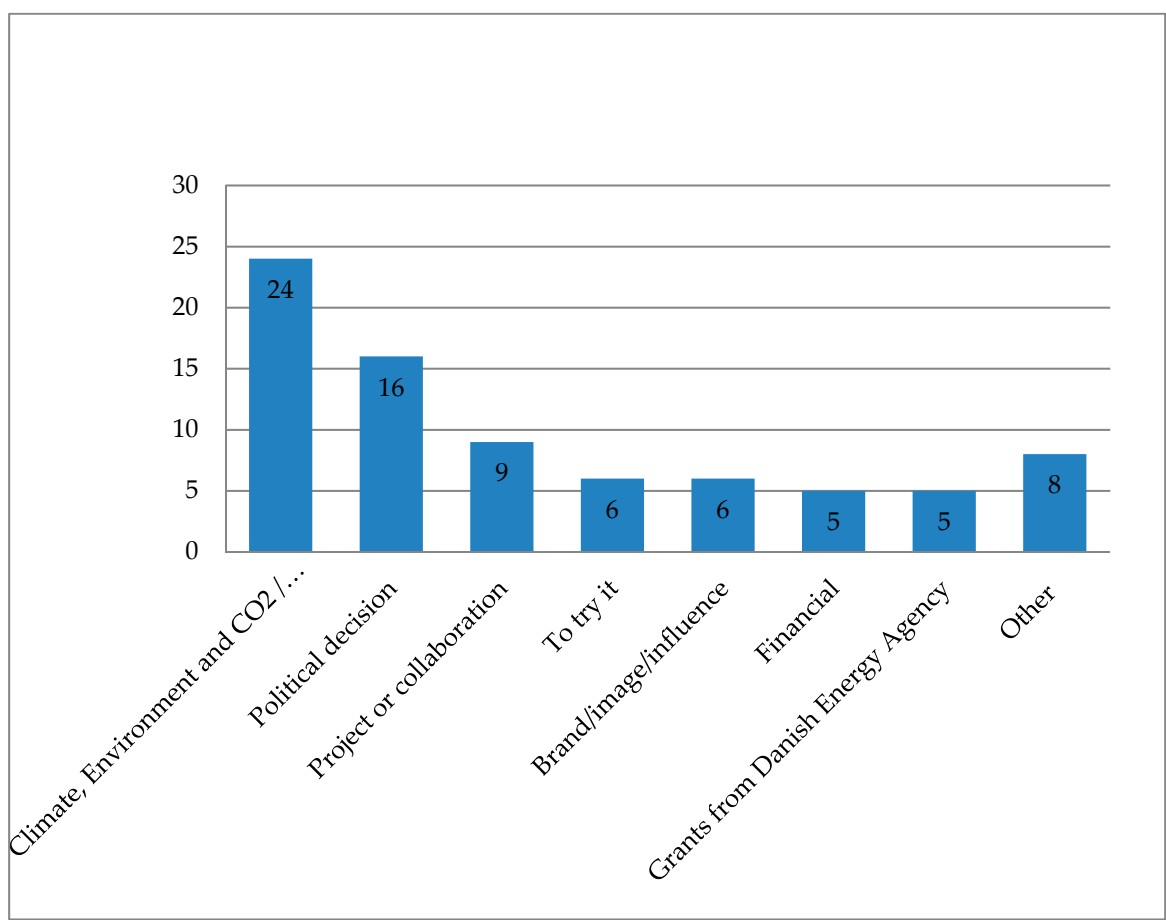

**Figure 3.** Factors that influenced municipalities' initial adoption of electric vehicles (from Question 4 of the survey: 'Why did your municipality decide to adopt electric vehicles?').

There were 69 total projects, which incentivised the municipalities buying or leasing PEVs and over half (35) were promoted or funded by the government. The survey question, 'Why did your municipality decide to adopt electric vehicles', was difficult to define in terms of question/answer as many of the municipalities had special agreements with certain car companies to try their cars for a certain period. Many of the municipalities highlighted the role of being involved in governmental projects as one of the key facilitators of procuring PEVs.

The projects varied in size and scope; however, they all provided some level of financial incentive for the municipality. Figure 4 shows the regional proportion of projects and the year during which they were implemented. Between 2013 and 2015, there was an overall peak of projects in Denmark, in particular the Capital Region and Central Denmark. In some of the interviews with the smaller municipalities (Frederikssund), the respondent mentioned the municipality might not have procured PEVs had it not been for its involvement in a project that had incentivized it [27].

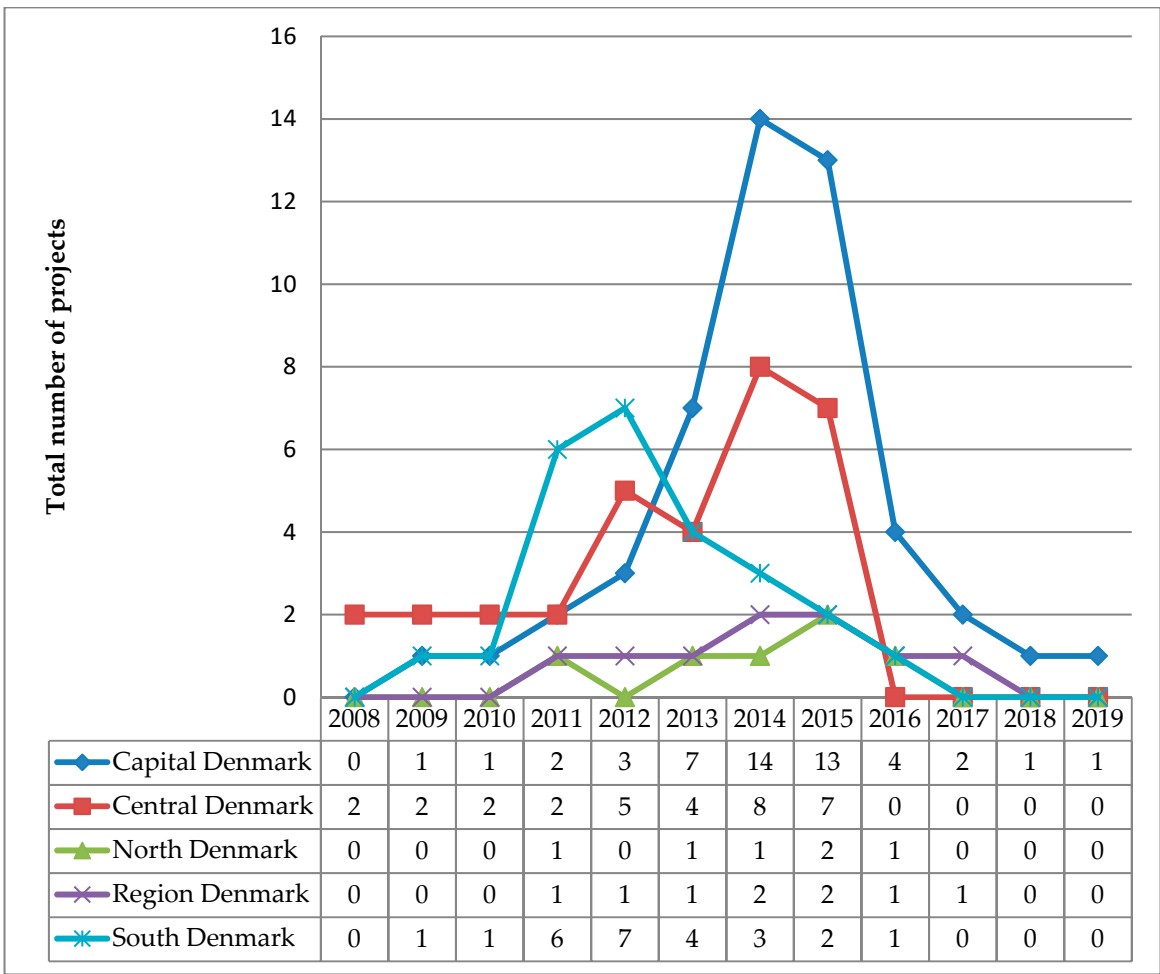

| | 2008 | 2009 | 2010 | 2011 | 2012 | 2013 | 2014 | 2015 | 2016 | 2017 | 2018 | 2019 |
|---|---|---|---|---|---|---|---|---|---|---|---|---|
| Capital Denmark | 0 | 1 | 1 | 2 | 3 | 7 | 14 | 13 | 4 | 2 | 1 | 1 |
| Central Denmark | 2 | 2 | 2 | 2 | 5 | 4 | 8 | 7 | 0 | 0 | 0 | 0 |
| North Denmark | 0 | 0 | 0 | 1 | 0 | 1 | 1 | 2 | 1 | 0 | 0 | 0 |
| Region Denmark | 0 | 0 | 0 | 1 | 1 | 1 | 2 | 2 | 1 | 1 | 0 | 0 |
| South Denmark | 0 | 1 | 1 | 6 | 7 | 4 | 3 | 2 | 1 | 0 | 0 | 0 |

**Figure 4.** Chart to show the percentage of all electric vehicle (EV) projects by Danish region.

### 6.3. Infrastructure, Charging, and Range

One of the keys issues evident from the surveys and interviews was range. Many municipalities highlighted a concern that the range was not large enough. The negative comments from the survey show a number of interesting trends (Figure 5), namely the most common complaint is 'problems in the winter'. Whilst science shows that in colder weather, the range is reduced slightly (due to running heaters), it is not significantly impacted [8]. Whilst this can be a negative, recommendations of how to plan for cold weather usage would help (such as pre heating the car whilst it is still plugged in). One of the common negative comments included running the EV outside of the municipality, where the user would often be unsure of where they could charge the car [27].

The interview responses to the range supported the findings from the survey, insofar as the respondents were uncertain about the range and felt this was one of the major drawbacks. This uncertainty is not uncommon when dealing with new technology. There has been limited research into how actual users of PEVs respond to range and how they feel about. Response from the interviews highlighted that it took some time for the drivers to get used to, and over time they found the range was sufficient for their needs and the only aspect was having to 'plan more in advance' for the journeys they would be taking. Initially, the range was more of a 'psychological' barrier according to the interviewees rather than a practical one. The more the driver of the PEV used the car, the better they understood the range and its limitations. Those municipalities, which were satisfied with the range, expressed in the survey that the range was sufficient for their needs and for the size of the municipality.

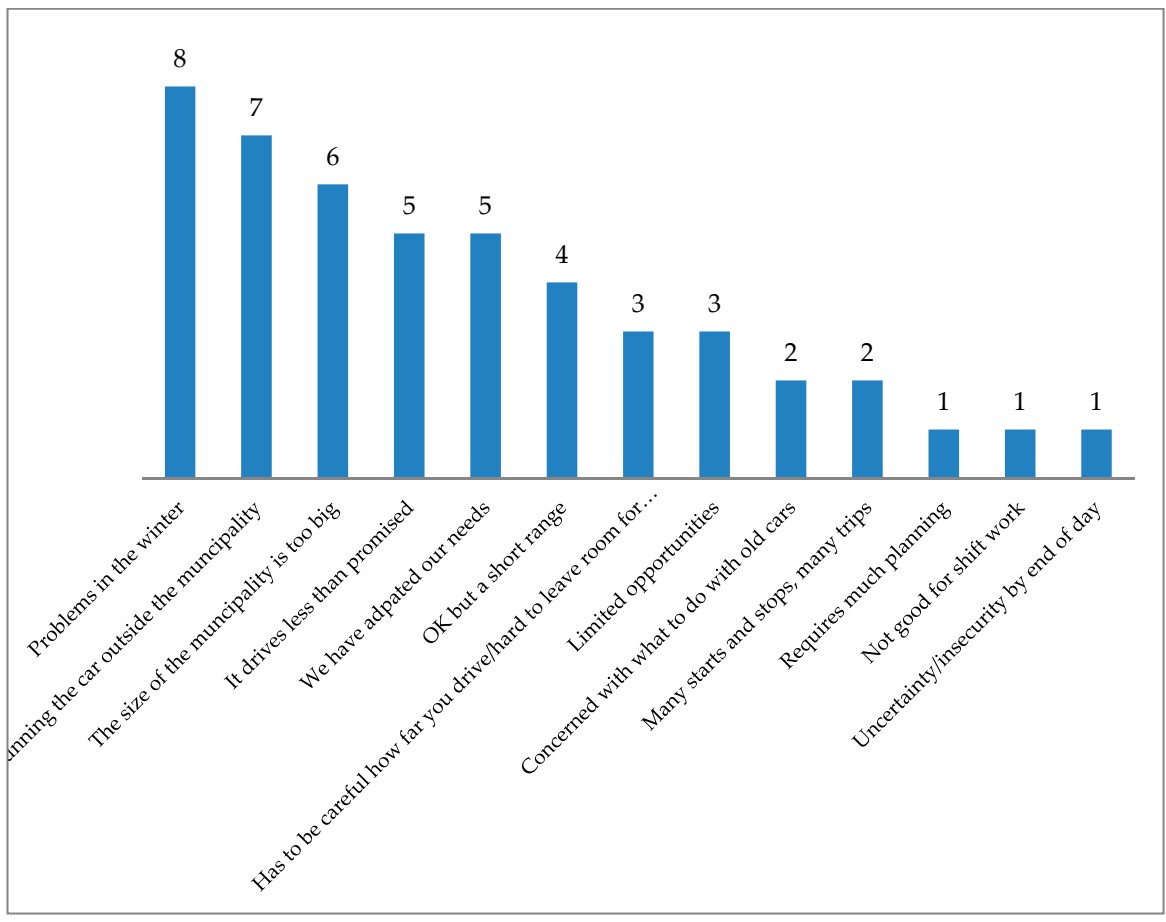

**Figure 5.** Graph depicting the most common negative comments about the range of EVs.

The range is intrinsically linked to the location and knowledge of charging stations. Questions 12 and 13 of the survey asked about the satisfaction with the level of charging infrastructure in their municipality. The survey asked them to rate the charging infrastructure out of five (five being the most satisfied). The average was 2.4, indicating an average satisfaction. All four interviewees explained that the users of the PEVs would only usually use the charging stations at the municipality and generally not use other charging stations. The majority of departments using the PEVs were 'home help' assistance, where the maximum stop time around the municipality would be approximately 45–60 min. Residences would often not be near a charging station and if they were, there was not enough time to fully charge the vehicle.

*6.4. Future of PEVs for Danish Municipalities*

Overall results from both the survey and interview suggest that the future of PEVs in their municipality is uncertain. When asked whether the municipality would expand their fleet of PEVs in the next 3 to 5 years, the majority of survey respondents replied positively. However, a large proportion replied that they did not know. The interview respondents supported this finding by explaining that whilst they would be keen to procure more PEVs in the future, they did not know what the political or economic forecast would be. Whilst it is the decision of the municipality to procure PEVs, there is a large amount of political persuasiveness from central government agencies. All the interviewee respondents expressed that the decision to procure more PEVs for the future would be a 'higher up managerial decision'. Whilst the fleet managers in the municipalities have a level of decision-making autonomy, the overall budget and incentives are from senior employees in the municipality. Table 1 highlights some of the exemplary quotes from the interviews.

**Table 1.** Exemplary quotes (interviews) from textual categories.

| Textual Category | Exemplary Quote |
| --- | --- |
| Reasons for procuring PEV | We wanted to be one of the leading municipalities in Denmark with regards to electric vehicles. We knew it would be expensive at the beginning by in the future it would be cheaper. |
| Financial incentives | We were given some help from the government with paying for the electric vehicles but we were in competition with other municipalities for this money. |
| Charging infrastructure | There are adequate charging stations but we have to plan for longer trips around the municipality and further. |
| Problems with cars | We were unsure about the range, and were hesitant about taking it outside the main towns where we knew where the charging stations were. |
| Who uses PEVs | We mainly use the cars for our 'home help' service for the elderly and disabled, they are small cars and do short journeys around the municipality. |
| Future | As a municipality we are uncertain about the future of electric vehicles, we would like to invest more money into buying more, but we don't know how much this will be. |

## 7. Local Government and the Procurement of Electric Vehicles: Policy Recommendations and Future Research

This paper has attempted to assess the decision-making pathway of how local municipalities across Denmark have attempted to procure electric vehicles as part of their wider fleet vehicle network. The questionnaires and interviews give valuable insight into the factors that have influenced how the municipalities procured the EVs. Central government plays a significant role in providing financial incentives for the local municipalities and as with Sierzchula's findings (Sierzchula 2014) [3], an enthusiastic fleet manager was often the driving force behind expanding a municipality's EV fleet. The data collection and analysis provided a solid foundation for the understanding and advancement of the processes and incentives for acquiring EVs.

Whilst incentives and enthusiastic individuals have been the main drivers behind procurement, the results suggest that the longevity of this is waning and without these incentives, the future of EVs in local municipalities is uncertain. More research needs to be done on the type of incentives offered and how these can be incorporated into long-term decision-making processes to ensure that that the number of EVs continues to expand.

## 8. Conclusions

This study used empirical survey and interview data to identify factors that influence the procurement of PEVs in a local governmental setting as well as understanding the decision-making pathways associated with the procurement of PEVs. Whilst this was a simple analytical approach, this research aimed to contribute to theory and provide a better understanding of early adopters of PEVs [28–31].

It was clear from the survey and interview that there were four main aspects associated with the municipalities:

- Reasons for procurement;
- decision making pathways of procurement;
- advantages and disadvantages of PEVs; and
- future of procurement of PEVs.

Referring to the research question, when identifying factors that influenced the procurement of PEVs, the most commonly stated factor was environmental followed by economic and political.

It was evident from the survey that each municipality manages the decision making of PEVs differently. The larger municipalities are often more proactive and have a larger number of PEVs, corresponding to a larger budget. The act of finding the most appropriate person in the municipality to complete the survey was challenging.

As early adopters, Danish municipalities are making progress, and many employees will chose to drive a PEV over a petrol or diesel car. The PEVs serve a role, and whilst they are not used for long trips outside the municipality, they are primarily used for short urban trips during the day. Therefore, there is high visibility to citizens, who ask questions and often want to know more about PEVs. It was unclear from the study, however, what the employees' own attitudes were on owning a PEV as a private car. It was indicated that the major barrier was price and if a PEV would be bought, it would be as a second car in the household. Municipality employees were also unsure as to the resale opportunities of PEVs with fast rate of technological development.

PEVs as fleet vehicles are an important step to the longer-term widespread adoption of PEVs. In a country, such as Denmark, where the vehicle taxes are very high for owning vehicles, including PEVs, this can dissuade citizens to invest in this alternative fuel market. Denmark's sustainable transport history has largely been associated with developing bicycling infrastructure and this has been very successful. Only time and technology will tell if local municipalities will provide the stepping stone for widespread adoption.

**Funding:** This research did not receive any specific grant from funding agencies in the public, commercial, or not-for-profit sectors.

**Conflicts of Interest:** The author declares no conflict of interest.

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
