# Peer review of "Electric vehicles in Danish Municipalities: An Understanding of Motivations, Barriers, and the Future of Sustainable Mobility"

_vehicles, doi:10.3390/vehicles1010004_

Reviewer 1 Report and Author's Response

Abstract: Good and concise - but perhaps surprisingly, does not refer to fleets / fleets users, nor do the Keywords
Response: I have added fleet and fleet cars to the abstract and keywords.

1. Introduction:

Line 23-24: It is rather unclear - as written here - if the new Danish government had cancelled or contunued the policies regarding taxes on EVs, "money saving" for whom ? consumers (which would mean lowering taxes / reversing previous tax increases or for the government (tax revenue increase)
Response: I have adjusted this sentence to hopefully make better sense. The Danish government has essentially an ‘on again off again’ approach to tax on electric vehicles. Originally it was almost nothing which cost the government a lot of money hence then increasing the tax on PEVs to earn some more money for the government.

Lines 34 and 37: which type of instruments do Danish municipalities have that are relevant to EV take-up ?
Response: The municipality vehicles are often seen driving around the local area and therefore have high visual presence in the area. I have outlined this in the paper.

Lines 57 / 58 needs to be re-written. Both PEVs and HEVs are EVs, and many BEVs are fact also PEVs. Better to distinguish between full electric vehicles and hybrid electric vehicles. Full (battery only) PEVs then as a sub-set and focus here.
Response: I have changed this in the text. The study focuses on just full PEVs.

Line 60-65: The focus of the paper appears to be here both BY Danish Municipalities and IN those muncipalities - needs to be clearer in Abstract and also earlier part of Introduction.
Response: The focus on the paper is understanding what factors influence the uptake of PEVs in the municipalities. This could come from a number of different areas, government policy, experience etc. I have altered this in the text accordingly.

2. Research Questions
Fine overall, but:
Line 73-75: these points must be supported by references
Response: These have now been referenced.

3. State of Knowledge
Overall good, if not fully comprehensive
Line 100 - needs the actual references in there
Response: I have added the reference here.

Line 109-110 - needs grammar / English phrasing correction
Response: I have re written these lines.

Line 116: what are 'ECVs' ?
Response: I have changed this, it should be PEVs.

4. Research Design
Overall good, but:
Line 157: PEVs
Response: Corrected.

Line 172 'using the EV so the information'
Response: Corrected.

Line 173: 'it was often the case'
Response: Rewritten the sentence.

5. Analysis
overall ok, though I do NOT see any of the inferential statistics promised in the Research Design section above !
Line 216: PEVs
Response: Changed.

Fig. 2 - what is 'ooo' on there ?
Response: This should be NA, some people filled out this as NA.

Line 240: environmental reasons. Is it possible to differentiate this into GreenHouse Gas (or more narrowly CO2 = decarbonisation of transport) emissions and urban air pollution (public health) ?
Response: I have altered this in the text.

Lines 246-247 - can that be made clearer ?
Response: It was often ambiguous from the interviewee what political decisions were made as they had no control of this.

Line 251 - what is 'this survey question'
Response: Re written.

Fig 4 caption / Line 258: I don't see where it does do Percentages anywhere, all absolute data
Response: I have changed this to total not percentages.

Lines 267-268: needs a reference
Response: This has been referenced.

Line 269: 'to get used to the EV'
Response: This has been re written.

Line 283: 'those municipalities which were happy
Response: Rewritten this sentence.

Line 303: 'in the municipalities have a level'
Response: This has been re written

section 7.
Rewrite lines 317-319 - pretty contradictory
Response: I have re written these few sentences.

Agree on recommdation for future research.

8. Conclusion
Line 337: 'when identifying factors which influenced'
Response: This has been reworded.

Line 339-345: shoudl this not be n the introduction / background earlier ?
Response: This has been moved to the introduction.

Line 349: incomplete sentence
Response: Deleted

Line 350: For the very first time, employees are metinoned, and also having the option to take or not to take up in terms of using EVs - before that, it was only ever about fleet managers' decisions
Response: I have reassessed this.

Line 354: employees'
Line 362. 'very successful. Only time'
Response: Re written

Lines 364-365: needs to do into the Funding section below
Response: Done

Reviewer 2 Report and Author's Response

Needs consistency in the spelling of acronyms (e.g., PEVs, PEV's, and PEVS all appear).
Response: This has been done. I have made all the acronyms are the same.

Line 39: It is unclear whether the amount of charging infrastructure has a significant effect on PEV adoption as stated, or vice-versa. It may only be a correlation, and not a causal effect. Should provide a reference to the stated claim.
Response: Reference inserted here.

Line 54: The statement "the crucial factor determining the success or failure of a new technology is users' acceptance" is tautological.
Response: This is somewhat true. I have reworded this sentence.

Line 58:  There seems to be a misstatement with "hybrid vehicles (Battery Electric Vehicles, BEVs)"
Response: This has been reworded.

Line 191:  "... where able to write"  should be "... were able to write"
Response: This has been reworded

Line 267:  The effect of cold weather on range is debatable, and many reports have shown an opposite conclusion.  Need to provide a reference to support the claim made here.
Response: A reference has been added.

Line 293:  45-60 minutes
Response: Re written

Line 308d:   should be "... would be expensive at the beginning BUT in the future..."
Response: Changed.

Round  2

Reviewer 1 Report and Author's Response

Line 24 should read: ... (from the government) ...
Line 26: ... what is clear that the annual number of PEVs newly registered has fallen ...
Response:
I have changed the Line 24 and also line 26.
Thank you for your help working through this paper.